# LightningDrag: Lightning Fast and Accurate Drag-based Image Editing Emerging from Videos

Yujun Shi [* 1]   Jun Hao Liew [* 2]   Hanshu Yan [2]   Vincent Y. F. Tan [1]   Jiashi Feng [2]

## Abstract

Accuracy and speed are critical in image editing tasks. Pan et al. introduced a drag-based framework using Generative Adversarial Networks, and subsequent studies have leveraged large-scale diffusion models. However, these methods often require over a minute per edit and exhibit low success rates. We present LIGHTNINGDRAG, which achieves high-quality drag-based editing in about one second on general images. By redefining drag-based editing as a conditional generation task, we eliminate the need for time-consuming latent optimization or gradient-based guidance, achieving high-quality editing in <1s. Our model is trained on large-scale paired video frames, capturing diverse motion (object translations, pose shifts, zooming, *etc.*) to significantly improve accuracy and consistency. Despite being trained only on videos, our model generalizes to local deformations beyond the training data (*e.g.*, lengthening hair, twisting rainbows). Extensive evaluations confirm the superiority of our approach. The code and model are available at https://github.com/magic-research/LightningDrag.

## 1. Introduction

Image editing using generative models (Roich et al., 2022; Endo, 2022; Hertz et al., 2022; Mokady et al., 2023; Kawar et al., 2023; Parmar et al., 2023) has received considerable attention in recent years. However, many existing approaches lack the ability to conduct fine-grained spatial control. One landmark work attempting to achieve precise spatial image editing is DRAGGAN (Pan et al., 2023), which enables interactive point-based image manipulation on generative adversarial networks (GANs). Using their method, users initiate the editing process by selecting pairs of handle and target points on an image. Subsequently, the model executes semantically coherent edits by relocating the contents of the handle points to their corresponding targets. Moreover, users have the option to delineate editable regions using masks, preserving the integrity of the rest of the image. Building upon the foundation laid by Pan et al. (2023), subsequent works (Shi et al., 2023; Mou et al., 2023; Nie et al., 2023; Ling et al., 2023) have endeavored to extend this editing framework to large-scale pre-trained diffusion models (Rombach et al., 2022), aiming to further enhance its generality.

However, a common drawback among many methods within this framework is their lack of efficiency. Prior to editing a real image input by the user, DRAGGAN (Pan et al., 2023) requires applying a lengthy pivotal-tuning-inversion (Roich et al., 2022), a process that can consume up to 1 to 2 minutes. As for diffusion-based approaches such as DRAGDIFFUSION (Shi et al., 2023) and DRAGONDIFFUSION (Mou et al., 2023), they typically entail time-consuming operations such as latent-optimization or gradient-based guidance during editing. This inefficiency poses a significant barrier to practical deployment in real-world scenarios. What undermines the users' experiences even more is the low success rate of these methods. Since they are mostly zero-shot methods that lack explicit supervision to perform drag-based editing, they frequently struggle with either accurately moving semantic content from handle to target points or preserving the appearance and identity of the source image.

In this study, we introduce LIGHTNINGDRAG, a novel approach that achieves state-of-the-art drag-based editing while drastically reducing latency to less than 1 second, thereby making drag-based editing *highly practical* for deployment. To attain such rapid drag-based editing, we redefine the task as a specific form of conditional generation, where the source image and the user's drag instruction serve as conditions. Drawing inspiration from previous literature (Zhang, 2023; Xu et al., 2023; Hu et al., 2023; Chen et al., 2024; Alzayer et al., 2024), we leverage the reference-only

*Equal contribution   [1] National University of Singapore   [2] ByteDance Inc.. Correspondence to: Yujun Shi <shi.yujun@u.nus.edu>, Jun Hao Liew <junhao.liew@bytedance.com>, Jiashi Feng <jsh-feng@bytedance.com>.

*Proceedings of the 42$^{nd}$ International Conference on Machine Learning*, Vancouver, Canada. PMLR 267, 2025. Copyright 2025 by the author(s).

architecture to process source images for identity preservation. Additionally, to incorporate the user's drag instruction into the generation process, we encode the handle and target points into corresponding embeddings via a Point Embedding Network. These embeddings are then injected into self-attention modules of the backbone diffusion model to guide the generation process. This approach eliminates the need for repeatedly computing gradients on diffusion latents during inference, as had been done in previous methods, thereby significantly reducing latency to that of generating an image with diffusion models. As a conditional generation pipeline, our approach can be further accelerated by integrating off-the-shelf acceleration modules for diffusion models (*e.g.*, LCM-Lora (Luo et al., 2023b), PeRFlow (Yan et al., 2024)), a capability not possible with previous gradient-based methods.

To train our proposed model, we leverage video frames as our supervision signals. This choice is motivated by the fact that video motions inherently encapsulate transformations relevant to drag-based editing (Fig. 1), such as object translations, changing poses and orientations, zooming in and out, *etc*. Our training data is constructed from paired video frames. Firstly, we sample pixels that exhibit significant optical flow magnitude on the first frame as the handle points. Next, we employ CoTracker2 (Karaev et al., 2023) to identify the handle points' corresponding target points in the second frame. This procedure allows us to construct training pairs for our model on a large scale. By learning from such large-scale video frames, our approach significantly outperforms previous methods in terms of both accuracy and consistency.

Through comprehensive evaluation across images of diverse categories and styles, we showcase the substantial advantages of our approach in terms of both speed and quality. Our approach adeptly delivers editing results in accordance to the user's instructions with an imperceptible latency of less than 1 second. Furthermore, we delve into two key techniques, namely *source noise prior* and *point-following classifier-free guidance*, that enhance the accuracy and consistency of our pipeline during inference. Lastly, we explore two test-time strategies that users can employ to further refine drag-based editing results—*point augmentation* and *sequential dragging*.

One potential concern regarding our proposed approach is that certain editing instructions — particularly those involving local or nonrigid deformations (*e.g.*, lengthening hair, twisting rainbows) — are not explicitly represented in natural video motions. Intriguingly, however, our model generalizes remarkably well to such out-of-domain edits. We hypothesize this stems from the compositional richness of video-based motion cues, combined with our explicit point-following control mechanism. We analyze this gener-

alization capability in depth in Sec. 6.

Our contributions are summarized as follows: 1) We propose LIGHTNINGDRAG, a fast and accurate drag-based image editing framework trained entirely on videos, achieving sub-second latency without test-time optimization; 2) We develop test-time techniques, including a noise prior strategy and point-following classifier-free guidance, to further improve editing quality and control; 3) Our model exhibits strong out-of-domain generalization, successfully handling deformation instructions not seen during training. We provide both empirical evidence and conceptual insights to explain this behavior.

## 2. Related Works

**Generative image editing.** In light of the initial successes achieved by generative adversarial networks (GANs) in image generation (Goodfellow et al., 2014; Karras et al., 2019; 2020), a plethora of image editing techniques have emerged based on the GAN framework (Endo, 2022; Pan et al., 2023; Abdal et al., 2021; Leimkühler & Drettakis, 2021; Patashnik et al., 2021; Shen et al., 2020; Shen & Zhou, 2021; Tewari et al., 2020; Härkönen et al., 2020; Zhu et al., 2016; 2023). However, owing to the limited model capacity of GANs and the inherent challenges in inverting real images into GAN latents (Abdal et al., 2019; Creswell & Bharath, 2018; Lipton & Tripathi, 2017; Roich et al., 2022), the applicability of these methods is inevitably restricted. Recent advancements in large-scale text-to-image diffusion models (Rombach et al., 2022; Saharia et al., 2022) have spurred a surge of diffusion-based image editing methods (Hertz et al., 2022; Cao et al., 2023; Mao et al., 2023; Kawar et al., 2023; Parmar et al., 2023; Liew et al., 2022; Mou et al., 2023; Tumanyan et al., 2023; Brooks et al., 2023; Meng et al., 2021; Bar-Tal et al., 2022; Epstein et al., 2023). While many of these methods aim to manipulate images using textual prompts, conveying editing instructions through text presents its own set of challenges. Specifically, the prompt-based paradigms are often limited to alterations in high-level semantics or styles, lacking the precise spatial control.

**Point-based image editing.** Point-based image editing is a challenging task aiming to manipulate images in pixel-level precision. Traditional literature (Beier & Neely, 2023; Igarashi et al., 2005; Schaefer et al., 2006) have relied on non-parametric techniques. However, recent advancements driven by deep learning-based generative models, such as GANs, have propelled this field forward, with several notable contributions (Pan et al., 2023; Endo, 2022; Wang et al., 2022; Zhu et al., 2016). One notable work among these is Pan et al. (2023), which achieves impressive interactive point-based editing by optimizing GAN latent codes. Nonetheless, the applicability of this framework is limited by the inherent capacity constraints of GANs.

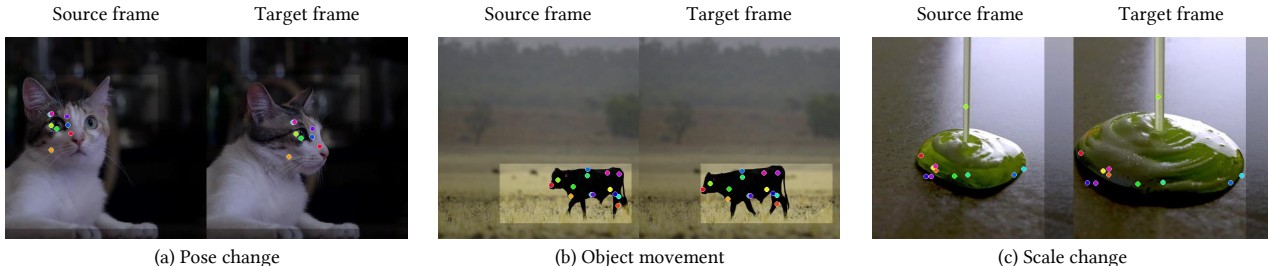

| Source frame | Target frame | Source frame | Target frame | Source frame | Target frame |

(a) Pose change          (b) Object movement          (c) Scale change

*Figure 1.* **Samples of supervision pairs from videos.** Video motion contains various transformation cues such as pose change, object movement and scale change, which are useful for the model to learn how objects change and deform while avoiding appearance change.

In a bid to enhance its versatility, subsequent efforts have endeavored to extend the framework to large-scale diffusion models (Shi et al., 2023; Mou et al., 2023; Luo et al., 2023a; Geng & Owens, 2024; Cui et al., 2024; Liu et al., 2024). However, most of these works still rely on computationally intensive operations such as latent optimization or gradient-based guidance, necessitating repeated gradient computations on diffusion latents and rendering them impractical for real-world deployment. Different from these works, Nie et al. (2023) introduce a paradigm that obviates the need for gradient computation on diffusion latents. However, this paradigm still requires repeated diffusion-denoising operations, resulting in latencies comparable to gradient-based methods such as Shi et al. (2023). Recent works (Li et al., 2024; Chen et al., 2024; Shin et al., 2024) redefine drag-based editing as a generation task, reducing the editing latency to levels comparable to image generation with diffusion models. However, these methods remain domain-specific: Li et al. (2024) handles part-level movements in articulated objects, Chen et al. (2024) focuses on single-human clothing, and Shin et al. (2024) targets facial images. While some architectural components of our method are inspired by these prior efforts, we are the first to leverage general video data to construct large-scale supervision for drag-based editing. This enables our model to handle a wide range of motion patterns and deformations without relying on curated or domain-specific datasets. As a result, our approach uniquely generalizes to *general images*, making it suitable for practical deployment across a broad range of scenarios.

**Learning image editing from videos** Previous methods leveraging videos to aid in learning image editing typically sample two frames from a video to form a supervision pair. For example, Chen et al. (2023) utilize image pairs collected from videos for the same object to learn the appearance variations, thus improving their subject-composition pipeline. Alzayer et al. (2024) use video frames to supervise their proposed coarse-to-fine warping-based image editing pipeline. Luo et al. (2023a) train diffusion models on video data to improve drag-based editing performance. However, their approach still relies on time-consuming gradient-based

guidance and is trained on a limited dataset comprising only around 100 supervision pairs. In contrast to these works, we train a conditional generation pipeline on *large-scale* videos to perform fast and accurate *drag-based editing*.

## 3. Preliminaries

### 3.1. Latent Diffusion Models

Diffusion models (Sohl-Dickstein et al., 2015; Ho et al., 2020) demonstrate promising performance in visual synthesis. Rombach et al. (2022) proposed the latent diffusion model (LDM), which first maps a given image $x_0$ into a lower-dimensional space via a variational auto-encoder (VAE) (Kingma & Welling, 2013) to produce $z_0 = \mathcal{E}(x_0)$. Then, a diffusion model with parameters $\theta$ is used to approximate the distribution of $q(z_0)$ as the marginal $p_\theta(z_0)$ of the joint distribution between $z_0$ and a collection of latent random variables $z_{1:T} = (z_1, \ldots, z_T)$. Specifically,

$$p_\theta(z_0) = \int p_\theta(z_{0:T}) \, \mathrm{d}z_{1:T}, \qquad (1)$$

where $p_\theta(z_T)$ is a standard normal distribution and the transition kernels $p_\theta(z_{t-1}|z_t)$ of this Markov chain are all Gaussian conditioned on $z_t$. In our context, $z_0$ corresponds to the VAE latent of image samples given by users, $z_t$ is the latent after $t$ steps of the diffusion process. Specifically,

$$z_t = \sqrt{\bar{\alpha}_t} z_0 + \sqrt{1 - \bar{\alpha}_t} \epsilon, \qquad (2)$$

where $\epsilon \sim \mathcal{N}(0, \mathbf{I})$, and $\bar{\alpha}_t$ is the cumulative product of the noise coefficient $\alpha_t$ at each step.

Based on the framework of LDM, several powerful pretrained diffusion models have been released publicly, including the Stable Diffusion (SD) model (https://huggingface.co/stabilityai). In this work, our proposed pipeline is developed based on SD model.

## 4. Methodology

In this section, we formally present our LIGHTNINGDRAG approach. To start, we elaborate on the details of how we

construct supervision pairs for our model from videos in Sec. 4.1. Next, we describe the architecture design of our model in Sec. 4.2. Furthermore, we introduce some techniques we use during test-time to improve the editing results in Sec. 4.3. Finally, we introduce some strategies that users can employ to fix failure cases in Sec. 4.4.

### 4.1. Paired supervision from video data

One of the challenges we encounter is in collecting large-scale paired data for training the model, as obtaining user-annotated input-output pair on a large scale is nearly infeasible. In this work, we redirect our focus towards leveraging video data. Our key insight lies in the inherent motion captured within video, which naturally encompasses various transformations relevant to drag-based editing, including zooming in and out, changes in pose and orientation, *etc.* These dynamics offer valuable cues for the model to learn how objects undergo changes and deform.

We begin by curating videos with static camera movement, simulating drag-based editing where only local regions are manipulated while others remain static. Subsequently, we randomly sample two frames from these videos to serve as source $I_{\mathrm{src}}$ and target images $I_{\mathrm{tgt}}$, respectively. We resample another pair if the optical flow between the two images is too small. Next, we sample $N$ handle points $P_{\mathrm{hdl}}$ on $I_{\mathrm{src}}$ with a probability proportional to the optical flow strength, ensuring the selection of points with significant movement. We then employ CoTracker2 (Karaev et al., 2023), a state-of-the-art point tracking algorithm to extract the corresponding target points $P_{\mathrm{tgt}}$ in the target image $I_{\mathrm{tgt}}$. Finally, we adopt a similar approach as in Dai et al. (2023) to extract a binary mask $M$ highlighting the motion areas, indicating regions to be edited. Collectively, the tuple $(I_{\mathrm{src}}, I_{\mathrm{tgt}}, P_{\mathrm{hdl}}, P_{\mathrm{tgt}}, M)$ form the sample to train our LIGHTNINGDRAG. Examples showcasing the versatility of video data for training drag-based editing can be found in Fig. 1.

### 4.2. Architecture Design

We formulate the drag-based image editing task as a conditional generation problem, where the generated image needs to fulfill the following criteria: (1) unmasked area remains untouched; (2) image identity (*e.g.*, human face, texture, *etc.*) should be preserved after dragging; (3) the areas indicated by handle points should move to the target coordinates. To achieve this, our LIGHTNINGDRAG comprises three components: (1) an image inpainting backbone to enforce unmasked region remains identical; (2) an appearance encoder preserves identity of $I_{\mathrm{src}}$, (3) a point embedding network encodes the (handle, target) points pairs, accompanied by a point-following attention mechanism, which explicitly enables the model to follow the point instructions. The overall framework is depicted in Fig. 2. We next elaborate on

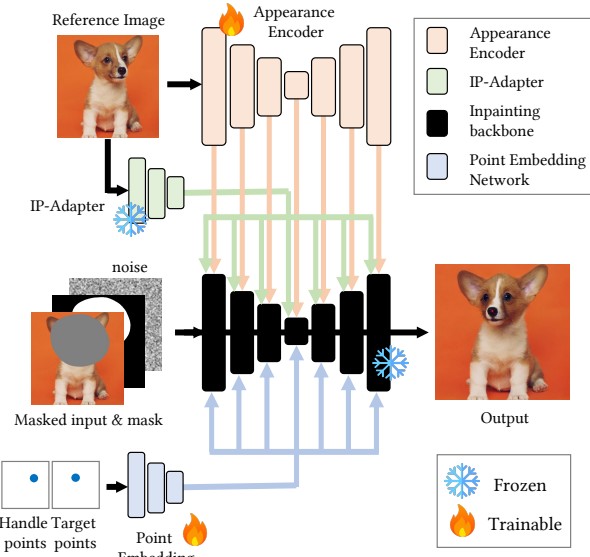

Figure 2. **The pipeline of** LIGHTNINGDRAG. Our LIGHTNING-DRAG consists of three components, including (1) an inpainting diffusion backbone to enforce unmasked regions remain untouched; (2) an Appearance Encoder for preserving the identity of the reference image; and (3) a Point Embedding Network to encode the (handle, target) points pairs.

each component in more details.

#### 4.2.1. INPAINTING BACKBONE.

We utilize the Stable Diffusion Inpainting U-Net as our backbone, which takes concatenation of the following as input: noise latents $z_t$, a binary mask $M$, and masked latents $M \odot z_0^{\mathrm{src}}$. It is worth noting that the inpainting backbone typically takes in a text prompt to indicate the inpainted content. However, in drag-based editing application, a text prompt is not only redundant as the image content is already provided by the source image, but also difficult for users to provide. Instead, we extract the image feature of the source image using IP-Adapter (Ye et al., 2023) and use an empty text prompt, freeing the users from this requirement.

#### 4.2.2. APPEARANCE ENCODER.

To maintain the identity of the reference image, we draw inspiration from recent works on ID-consistent generation, such as Xu et al. (2023); Hu et al. (2023); Chen et al. (2024). Specifically, we employ the reference-only architecture (Zhang, 2023) to process the source image. Unlike CLIP image encoder (Radford et al., 2021) which can only guarantee the overall colors and semantics, the reference-only approach has demonstrated efficacy in preserving fine-grained details of the reference image. Inherited from the weights of a pre-trained text-to-image U-Net diffusion model, our Appearance Encoder takes the reference latents $z_0^{\mathrm{src}}$ as in-

put. It extracts the reference feature maps from the self-attention layers, which are subsequently used to guide the self-attention process in the denoising backbone. The self-attention in the backbone is thus defined as follows:

$$\text{Attn}(Q, K, V, K_{\text{ref}}, V_{\text{ref}}) = \text{SM}\Big(\frac{Q[K, K_{\text{ref}}]^\top}{\sqrt{d}}\Big)[V, V_{\text{ref}}], \tag{3}$$

where $K_{\text{ref}}$ and $V_{\text{ref}}$ denote the keys and values extracted from the reference features, $[\cdot, \cdot]$ denotes the concatenation operator, and SM is the softmax function. Following prior works (Xu et al., 2023), we use clean reference latents as inputs to the Appearance Encoder (as opposed to noised latents used in original reference-only model (Zhang, 2023)). As a result, unlike backbone UNet that requires multiple denoising steps, the Apppearance Encoder only needs to extract features once throughout the entire editing process, which improves the model inference efficiency.

### 4.2.3. POINT EMBEDDING ATTENTION.

Given the user-specified handle and target points, we first convert them into a handle and a target point map that is of the same resolution of the input image. Specifically, we randomly assign each pair of handle and target points with an integer number $k \in \{1, 2, \ldots, N\}$, where $N$ is the maximum allowed points. Then, we put the integer $k$ to the pixel location on the point map given coordinates specified by handle and target points. The rest of the pixel locations on handle and target point maps are with value 0.

Once obtaining the handle and target point maps, we encode them into embedding via a point embedding network, which is composed of 12 layers of convolution and SiLU activation. This network outputs embedding at four different resolutions, corresponding to the four different resolutions of SD UNet activation maps. To enable the model to follow point instructions effectively, we draw inspiration from Chen et al. (2024) and introduce a point-following mechanism into Eqn. 3, resulting in the following formulation:

$$\text{Attn}(Q, K, V, K_{\text{ref}}, V_{\text{ref}}, E_{\text{hdl}}, E_{\text{tgt}})$$
$$= \text{SM}\Big(\frac{(Q + E_{\text{tgt}})[K + E_{\text{tgt}}, K_{\text{ref}} + E_{\text{hdl}}]^\top}{\sqrt{d}}\Big)[V, V_{\text{ref}}] \tag{4}$$

where $E_{\text{hdl}}$ and $E_{\text{tgt}}$ are embeddings of handle and target point maps, respectively. In this way, we explicitly strengthen the similarity between the target points of the generated images and the handle points of the user input image, facilitating learning of drag-based editing.

### 4.3. Test-time Techniques to Improve Editing Results

#### 4.3.1. NOISE PRIOR

We have observed that directly using randomly initialized noise latents for generation sometimes yields unstable re-

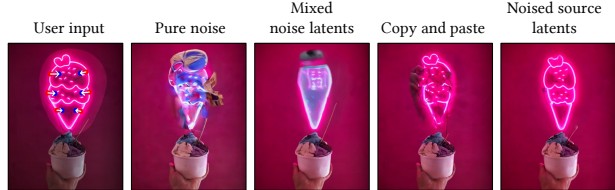

*Figure 3.* **Different strategies for constructing the noise prior.** We find that the "noise source latents" strategy produces the best results. Image credit (source image): Pexels

sults, as depicted in Fig. 3. This instability may stem from the discrepancy between the initial noise during training and testing of diffusion models, as discussed in prior works (Lin et al., 2024a;b). In contrast to text-to-image generation, where obtaining a suitable initial noise prior is challenging, our task allows for a more accurate initialization of the noise prior by adding noise to the VAE latent of the source image. This technique enables us to narrow the gap between training and testing, resulting in more stable outcomes.

We ablate three strategies to construct the noise prior:

- *Noised source latents* directly adds noise on the source image latent with Eqn. (2) to the terminal diffusion time-step ($t = 999$).
- *Mixed noise latents* re-initializes the mask region of the noised source latents with pure Gaussian noise for potentially greater editing flexibility.
- *Copy and paste noise latents* borrows the "copy and paste" approach of Nie et al. (2023) along with the handle and target points to form the initial noise prior.

We report results using two metrics: Mean Distance (MD↓), which measures how accurately the handle points are moved to the target positions, and Image Fidelity (IF↑), calculated as 1-LPIPS, which reflects how well the edited image preserves the original appearance. As shown in Tab. 1, the "copy and paste" and "noise source latent" strategies yield the best overall performance, with the latter slightly outperforming others in IF. However, since "copy and paste" involves much more complicated implementation, we select "Noise Source Latents" as our default strategy in the spirit of simplicity. Moreover, the numerical results presented here also align well with the visual illustration in Fig. 3.

|  | Pure Noise | Mixed Noise Latents | Copy and Paste | Noise Source Latents |
|---|---|---|---|---|
| MD(↓) | 19.76 | 19.01 | 18.53 | 18.95 |
| IF(↑) | 0.887 | 0.886 | 0.887 | 0.890 |

*Table 1.* Quantitative ablation on different noise prior strategies during inference.

### 4.3.2. POINT-FOLLOWING CLASSIFIER-FREE GUIDANCE

To further improve the model's capability to follow the point instruction during inference, we implement the following *point-following classifier-free guidance* (PF-CFG) to strengthen the effects of given (handle, target) points pairs:

$$
\tilde{\epsilon}_\theta(z_t, c_{\mathrm{appr}}, c_{\mathrm{points}}) = \epsilon_\theta(z_t, c_{\mathrm{appr}}, \emptyset)
$$
$$
+ \omega(t)\big(\epsilon_\theta(z_t, c_{\mathrm{appr}}, c_{\mathrm{points}}) - \epsilon_\theta(z_t, c_{\mathrm{appr}}, \emptyset)\big),
\tag{5}
$$

where $\omega(t)$ is the time-dependent CFG scale, $c_{\mathrm{appr}}$ denotes the source image condition encoded by appearance encoder, and $c_{\mathrm{points}}$ denotes the condition of handle and target points. To be more specific, when computing $\epsilon_\theta(z_t, c_{\mathrm{appr}}, \emptyset)$, we use Eqn. (3) in all self-attention layers of the main backbone UNet. When computing $\epsilon_\theta(z_t, c_{\mathrm{appr}}, c_{\mathrm{points}})$, we employ Eqn. (4).

Most previous works involving diffusion models apply a fixed CFG scale across different denoising time-steps. However, recent literature (Kynkäänniemi et al., 2024; Wang et al., 2024) demonstrate the benefits of using a time-dependent CFG scale during denoising. In this work, we similarly find that a dynamic time-dependent CFG scale can help strike an appropriate balance between the accuracy of point-following and image quality of the results.

Denoting $\omega_{\mathrm{max}}$ as the maximum value of CFG, we explore the following CFG scale schedules:

- *No CFG:* $\omega(t) = 1$
- *Constant:* $\omega(t) = \omega_{\mathrm{max}}$.
- *Square:* $\omega(t) = \omega_{\mathrm{max}} \times (1 - (1 - t/1000)^2) + (1 - t/1000)^2$.
- *Linear:* $\omega(t) = \omega_{\mathrm{max}} \times t/1000 + (1 - t/1000)$.
- *Inverse square:* $\omega(t) = (\omega_{\mathrm{max}} - 1) \times (t/1000)^2 + 1$.

We compare these schedules in Fig. 4. As can be observed, without using our CFG, the model struggles to conduct successful drag-based editing. On the other hand, using CFG with a constant scale can successfully drag the handle points to the target, but the results may suffer from over-saturation. By using schedules that decay the CFG scale from $\omega_{\mathrm{max}}$ to 1.0 during the denoising process such as *Square*, *Linear*, and *Inverse square*, we achieve accurate drag-based editing while markedly improve the image quality. Among these decaying schedules, we find fast decaying strategy such as *Inverse square* achieves the best image quality, while slow decaying strategy such as *Linear* and Square still suffer from slight quality degradation (*e.g.,* over-saturation) on generated images.

To complement the qualitative observations above, we present a quantitative comparison of the different CFG schedules in Tab. 2. The results confirm our findings: constant CFG improves point accuracy (MD) but hurts image fidelity (IF), while dynamic schedules offer a better trade-off between control and realism. Among these schedulers, we select inverse square as our default CFG scale scheduler to attain the best overall appearance preservation while maintaining good point-following.

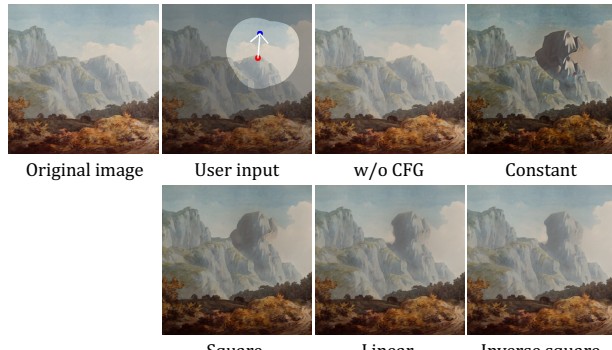

| Original image | User input | w/o CFG | Constant |
| Square | Linear | Inverse square | |

*Figure 4.* **Effects of different CFG scale schedules.** Our model struggles to conduct a successful drag when CFG is not used. Constant CFG often leads to over-saturation problem. Overall, fast decaying strategy (*Inverse square*) attains the best results.

| | w/o CFG | Constant | Square | Linear | Inv. Square |
|---|---|---|---|---|---|
| MD($\downarrow$) | 22.27 | 17.76 | 17.73 | 18.59 | 18.95 |
| IF($\uparrow$) | 0.9071 | 0.8486 | 0.8743 | 0.8829 | 0.8896 |

*Table 2.* Quantitative ablation of different CFG scale schedules.

### 4.4. "Drag engineering" to improve the editing

Inspired by the use of prompt engineering technique in Large Language Models (LLM) to obatin ideal answers, we find that some failure cases produced by our LIGHTNING-DRAG can also be mitigated by engineering the input drag instruction. Here, we introduce two strategies, namely *Point augmentation* and *Sequential dragging*, for users to consider when facing imperfect results with our model.

### 4.4.1. POINT AUGMENTATION

When the region specified by handle points fail to move to the target locations, augmenting the drag instruction with additional pairs of handle and target points has proven effective in improving results. Examples showcasing this augmentation are depicted in Fig. 5. It is evident that by incorporating more pairs of handle and target points, users' editing intentions can be more explicitly conveyed, resulting in better outcomes.

### 4.4.2. SEQUENTIAL DRAGGING

In cases where drag editing results are sub-optimal after one round of editing, users may opt to break down the drag instruction into multiple rounds and sequentially move

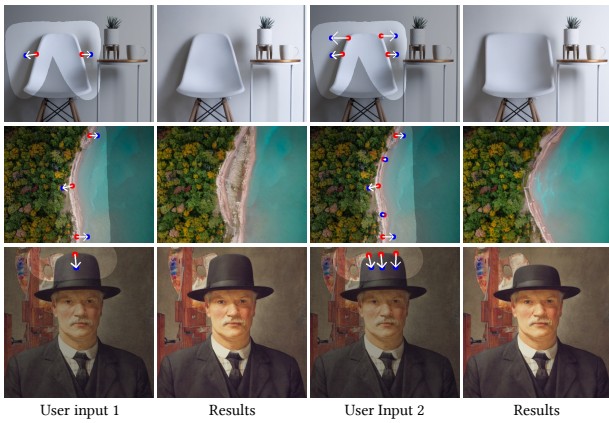

| User input 1 | Results | User Input 2 | Results |

*Figure 5.* **Point Augmentation.** Augmenting with additional pairs of handle and target points can better convey the user's editing intention, which often leads to better performance.

semantic contents from handle points to final targets. Examples illustrating how such sequential dragging can rectify certain failure cases are presented in Fig. 6. This strategy is facilitated by our model's exceptional ability to maintain the appearance and identity of the source image during editing. Without this capability, cumulative appearance shifts might occur, leading to undesired results. Additionally, given our model's negligible latency, employing sequential dragging does not significantly undermine user experience.

# 5. Experiments

## 5.1. Implementation details

**Network.** The base inpainting U-Net inherits the pre-trained weights from Stable Diffusion V1.5 inpainting model, whereas the Appearance Encoder is initialized from the pre-trained weights of Stable Diffusion V1.5. The Point Embedding Network is randomly initialized, except for the last convolution layer which is zero-initialized (Zhang et al., 2023) to ensure the model starts training as if no modification has been made.

**Training.** We sample 220k training samples from our internal video dataset to train our model. We set the learning rate to $5e-5$ with a batch size of 256. We freeze both the inpainting U-Net and IP-Adapter, training both Appearance Encoder and Point Embedding Network. During training, we randomly sample $[1, 20]$ points pairs. We randomly crop a square patch covering the sampled points and resize to $512 \times 512$.

**Inference.** We use DDIM (Song et al., 2021) sampling with 25 steps for inference by default. We found that our model is also compatible with recent diffusion acceleration techniques such as LCM-LoRA (Luo et al., 2023b) and PeRFlow (Yan et al., 2024) without additional training. When using LCM-LoRA or PeRFlow, we use 8 steps for sampling. We

| Method | IF (↑) | MD (↓) |
|---|---|---|
| DragGAN (Pan et al., 2023) | 0.696 | 57.155 |
| DragDiffusion (Shi et al., 2023) | 0.881 | 33.162 |
| DiffEditor (Mou et al., 2024) | 0.856 | 28.579 |
| Readout Guidance (Luo et al., 2023a) | 0.790 | 52.224 |
| SDEDrag (Nie et al., 2023) | **0.921** | 45.779 |
| LightningDrag (ours) | 0.885 | **18.62** |
| LightningDrag + LCM-LoRA (ours) | 0.890 | 18.95 |

*Table 3.* **Quantitative comparison on DragBench.** IF and MD denote Image Fidelity (1-LPIPS) and Mean Distance, respectively.

use guidance scale $\omega_{\max}$ of 3.0 and adopt an inverse square decay (Sec. 4.3.2) that gradually reduces the guidance scale to 1.0 over time to prevent over-saturation issue.

## 5.2. Evaluation on DragBench

We provide a quantitative assessment of our method on DragBench (Shi et al., 2023), comprising 205 samples with pre-defined drag points and masks. As is standard (Shi et al., 2023; Ling et al., 2023; Cui et al., 2024; Liu et al., 2024), we use the Image Fidelity (IF) and Mean Distance (MD) metrics for our analysis. IF is calculated as $1-$LPIPS (Zhang et al., 2018), while MD assesses the accuracy with which handle points are moved to their designated targets. An ideal drag-based editing method would achieve a low MD, indicating effective drag editing, coupled with a high IF, signifying robust appearance preservation.

Tab. 3 demonstrates the superiority of our LIGHTNING-DRAG in term of point following, as evidenced by its lowest MD. We also notice that our LIGHTNINGDRAG outperforms others in term of IF, except SDEDrag. However, further inspection reveals that SDEDrag often results in the undesired identity mapping (Fig. 7 row 1, 3 and 4), leading to its high IF. Additional qualitative results supporting this observation are presented in Fig. 8 in the appendix.

## 5.3. Time efficiency

Due to the elimination of test-time latent optimization or gradient-based guidance, our LIGHTNINGDRAG is extremely fast. We compare the time efficiency of LIGHT-NINGDRAG against the state-of-the-art methods for drag-based editing on general images. For fair comparisons, we extracted a subset of square images from DragBench (Shi et al., 2023), resulting in 67 images, and perform inference at a resolution of $512 \times 512$. We report the time cost on an NVIDIA A100 GPU in Tab. 4. We notice that the execution time of SDEDrag has a high variance. This is because its inference speed depends on the distance between the handle and target points. In contrast, our LIGHTNINGDRAG runs at a constant speed regardless of the dragging distance. Secondly, even without LCM-LoRA, our approach is already

**Single dragging**       **Sequential dragging**

Drag #1      Drag #2      Drag #3

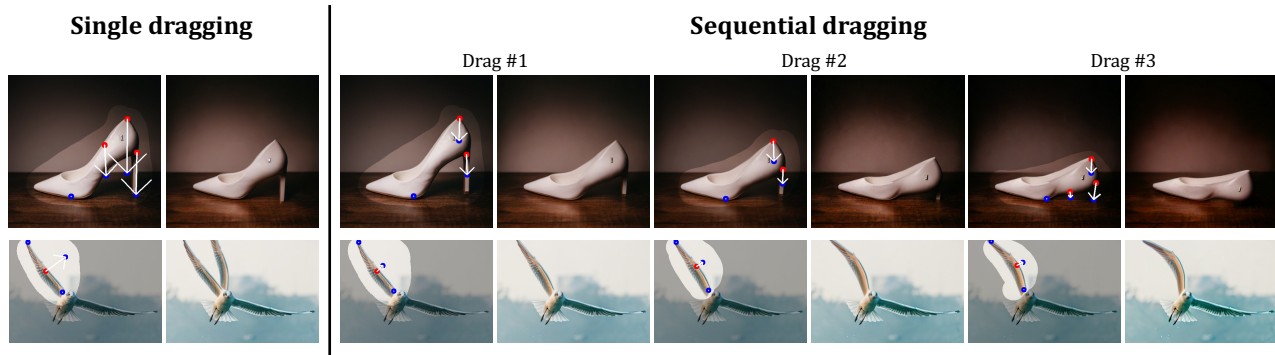

*Figure 6.* **Sequential dragging.** In cases when single dragging operation cannot attain the desired outcome, a simple workaround is to break the operation down into a sequence of shorter dragging trajectories. Image credit (source images): Pexels.

| Model | Time (s) |
|---|---|
| DragDiffusion (Shi et al., 2023) | $55.05_{\pm 3.70}$ |
| DiffEditor (Mou et al., 2024) | $9.47_{\pm 0.20}$ |
| Readout Guidance (Luo et al., 2023a) | $98.13_{\pm 0.52}$ |
| SDEDrag (Nie et al., 2023) | $61.78_{\pm 18.84}$ |
| LightningDrag (ours) | $2.06_{\pm 0.05}$ |
| LightningDrag + LCM-LoRA (ours) | $\mathbf{0.92}_{\pm 0.02}$ |

*Table 4.* **Time efficiency.** The reported time cost is obtained by running inference on $512 \times 512$ images sampled from DragBench (Shi et al., 2023) on a single NVIDIA A100 GPU.

an order of magnitude faster than most baselines, making it suitable for practical applications. Lastly, when combined with recent diffusion acceleration methods such as LCM-LoRA, our LightningDrag can be further accelerated, requiring only $< 1s$ for each dragging operation.

### 5.4. Qualitative results

**Comparisons with Prior Methods.** We compare our LIGHTNINGDRAG with prior methods in Fig. 7. We observe that DiffEditor and Readout Guidance often struggle to preserve reference identity (*e.g.*, 2nd and 3rd row), while DragDiffusion and SDEDrag sometimes fail to drag the regions-of-interest to the desired locations. In contrast, our LIGHTNINGDRAG effectively handles various dragging needs, such as pose change, object scaling, translation, local deformation, while preserving the source image appearance.

**Multi-round Dragging.** Our LIGHTNINGDRAG supports multi-round dragging, allowing users to iteratively refine edits based on previous outputs. Examples of multi-round dragging are shown in Fig. 9 in Appendix.

## 6. Discussion: Out-of-Domain Generalization

A key strength of LIGHTNINGDRAG lies in its surprising ability to generalize to out-of-domain editing instructions, particularly those involving non-rigid deformations (*e.g.*,

stretching hair, bending structures), even though such transformations are *not* explicitly observed in training videos.

We attribute this to two factors:

(1) **Compositional Generalization from Natural Video Cues.** Although our model is trained solely on videos without explicit dragging instructions, video motion contains rich non-rigid dynamics. For example, Fig. 1(c) shows syrup deforming as it accumulates — providing cues for transformations like stretching or warping. This suggests that the model learns a general motion prior from such examples, allowing it to extrapolate to unseen edits. We view this as a form of compositional generalization: the model internalizes structural transformations and recombines them during inference.

(2) **Explicit Control via PF-CFG.** Our Point-Following Classifier-Free Guidance (PF-CFG) is critical in guiding this generalization during inference. By enforcing alignment between semantic regions around handle and target points, PF-CFG improves the model's control and robustness. As evidenced in both Fig. 4 and our ablation study, this component enhances the reliability of drag instructions even under challenging or ambiguous cases.

Together, these insights suggest that our approach not only achieves speed and accuracy but also establishes a promising path toward more generalized and controllable image editing using natural video supervision.

## 7. Conclusion

We introduced LIGHTNINGDRAG, a practical approach for high-quality drag-based image editing in $\sim 1s$. By leveraging large-scale natural videos as a rich source of motion cues, our model learns how objects change and deform in diverse scenarios. Extensive experiments confirm LightningDrag surpasses prior methods in both speed and quality. We hope this work will inspire further research on controllable and precise image editing.

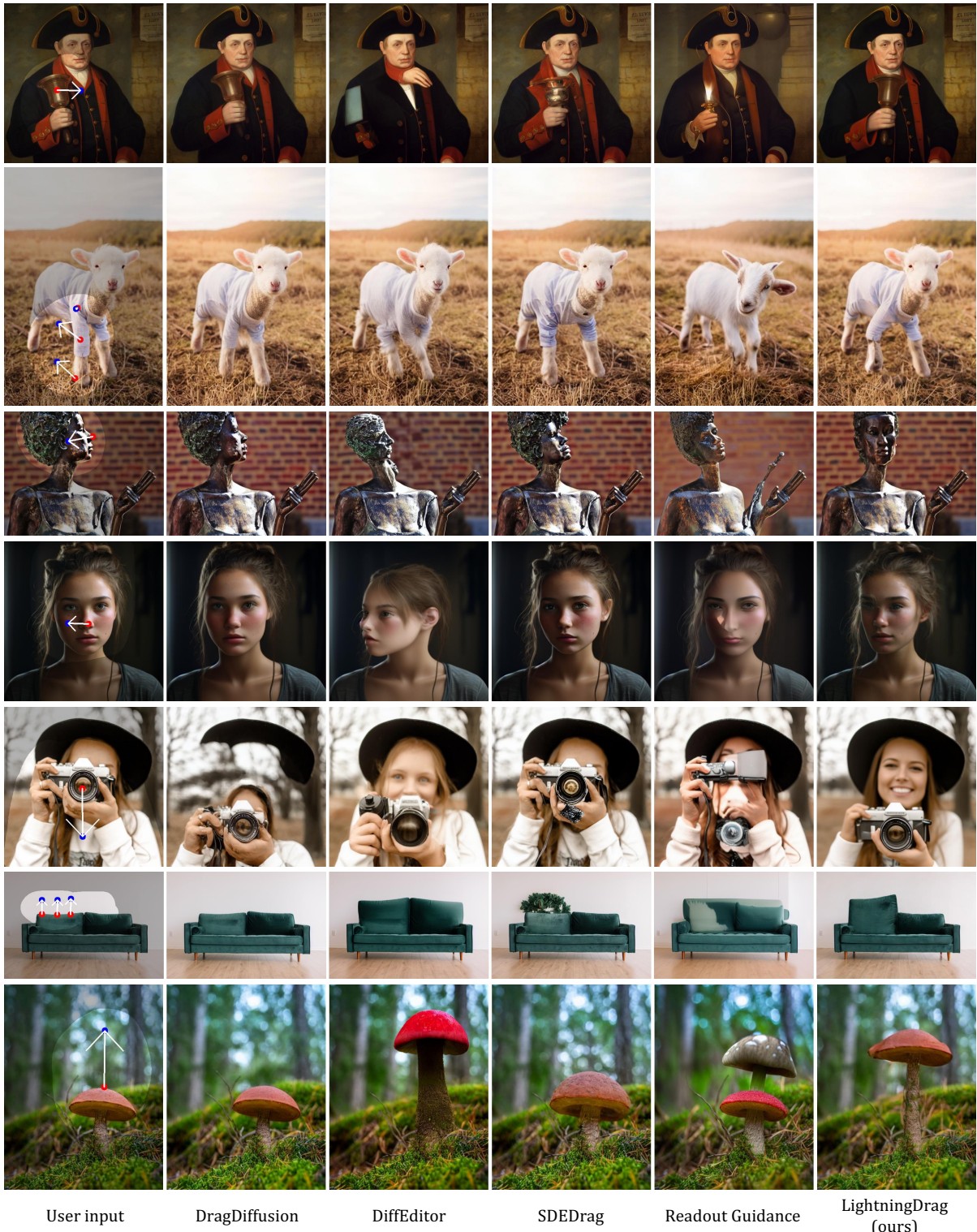

User input    DragDiffusion    DiffEditor    SDEDrag    Readout Guidance    LightningDrag (ours)

*Figure 7.* **Qualitative comparison on DragBench.** Our LightningDrag can handle various dragging instructions, such as pose change, scaling, translation *etc.* while preserving the object identity.

## Impact Statement

This paper introduces a fast and high-quality drag-based image editing approach, making advanced editing tools more accessible to a wider audience. Although this democratization of image manipulation has many positive applications, it also raises concerns about the potential misuse of generated content, such as creating deceptive or fake imagery. However, such risks are inherent in most research in visual content synthesis, including image and video generation.

## Acknowledgment

The authors would like to thank Zhongcong Xu, Zhijie Lin, Jianfeng Zhang, Zilong Huang, and the anonymous reviewers for their helpful discussion and feedback.

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

## A. Video Demo

Please see our project website for a video demonstration: https://lightning-drag.github.io/

## B. Additional Qualitative Results

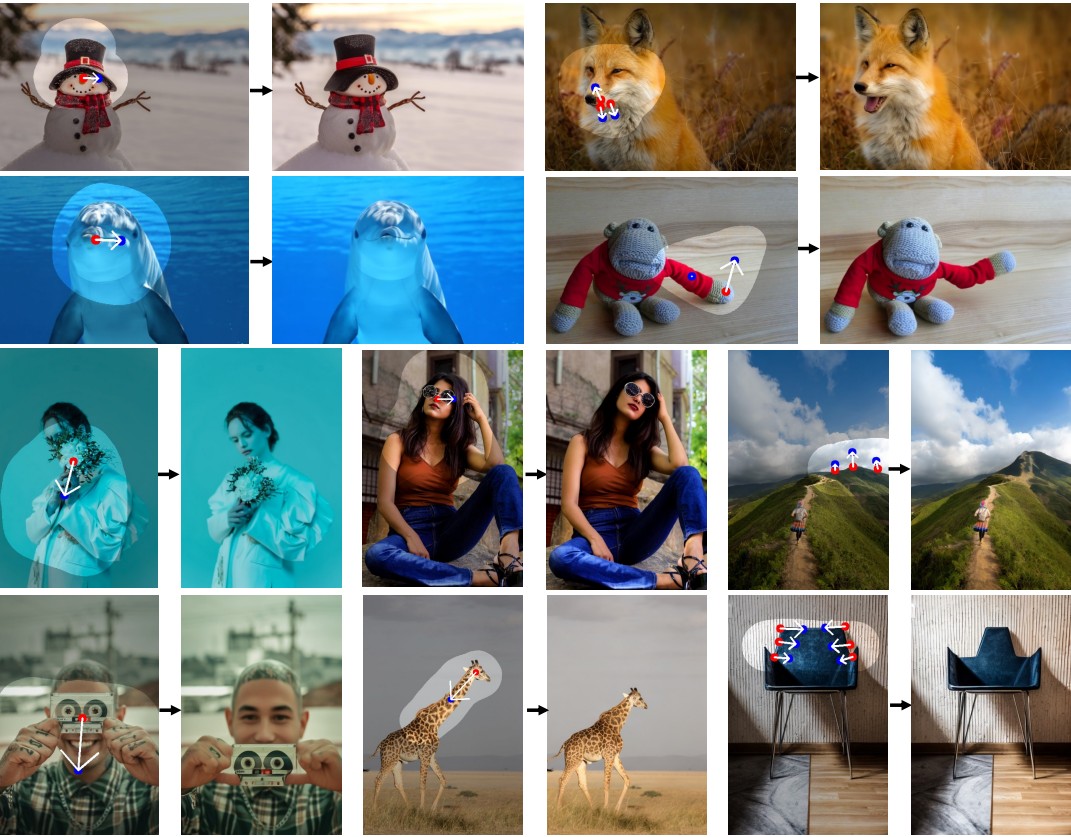

*Figure 8.* **Qualiative results of** LIGHTNINGDRAG. Image credit (source images): Pexels.

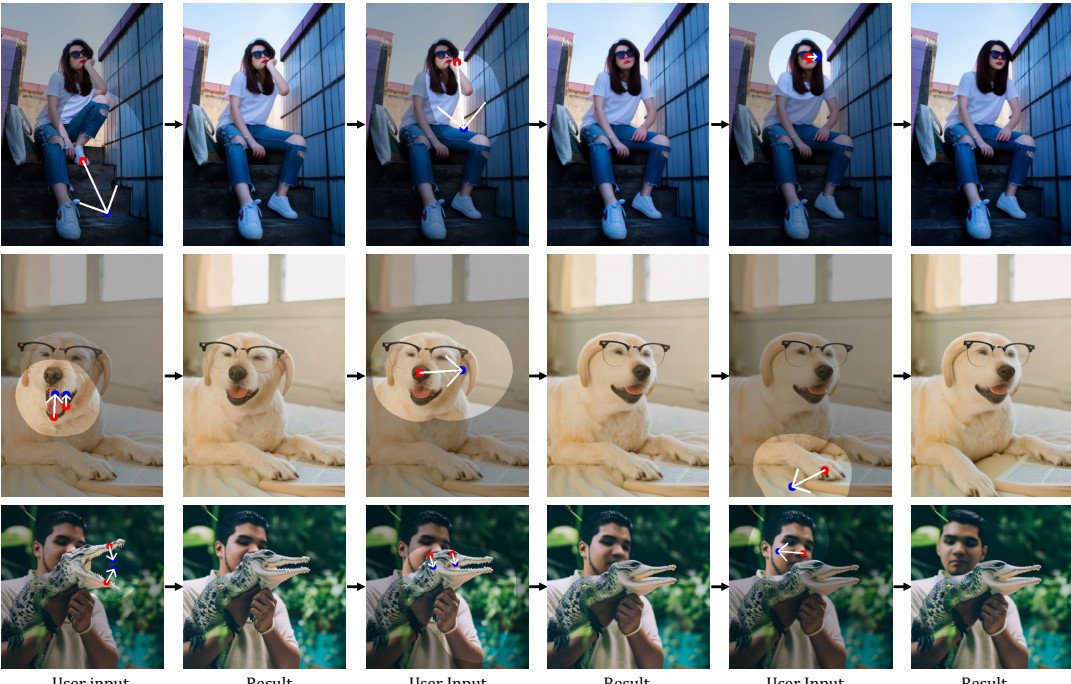

User input    Result    User Input    Result    User Input    Result

*Figure 9.* **Multi-round dragging.** Image credit (source images): Pexels.

