# OpenReview forum: "LightningDrag: Lightning Fast and Accurate Drag-based Image Editing Emerging from Videos"
_ICML.cc/2025/Conference — ICML 2025 poster_

### Official Review · Reviewer_qwJU · 2025-03-09

**Overall Recommendation:** 3

**Summary:**

This paper introduces LIGHTNINGDRAG, a high-speed, high-quality drag-based image editing framework that significantly outperforms existing methods in both efficiency and success rate. Unlike prior approaches that rely on Generative Adversarial Networks or large-scale diffusion models—often requiring over a minute per edit—LIGHTNINGDRAG completes edits in about one second. By redefining drag-based editing as a conditional generation task, the method eliminates the need for computationally expensive latent optimization or gradient-based guidance. Trained on large-scale paired video frames, the model captures diverse motion patterns, such as object translations, pose shifts, and zooming, leading to improved accuracy and consistency. Despite being trained only on videos, LIGHTNINGDRAG generalizes well to various local deformations beyond its training data, such as lengthening hair or twisting rainbows. Extensive evaluations demonstrate the superiority of this approach.

**Claims And Evidence:**

Yes

**Essential References Not Discussed:**

No.

**Experimental Designs Or Analyses:**

Yes

**Methods And Evaluation Criteria:**

Yes

**Other Comments Or Suggestions:**

1. It is suggested that the author add a section at the end of the introduction to summarize the contributions of this paper.

**Other Strengths And Weaknesses:**

1. The technical contribution is somewhat limited. The overall structure of LightningDrag is essentially the same as Wear-Any-Way[1], especially in the reference encoder and point embedding attention components.
2. The qualitative experiments compared to other methods are not very comprehensive. There are only seven comparative examples in the paper and supplementary materials.
3. The author claims in the paper P3L150 that 'We begin by curating videos with static camera movement,' but the example in Fig. 1(c) shows a forward-moving camera. Does the author keep such data during the filtering process?

[1] Wear-Any-Way: Manipulable Virtual Try-on via Sparse Correspondence Alignment. In ECCV 2024.

**Questions For Authors:**

The reviewer's primary concern is the novelty of this paper, and the score will be adjusted based on the author's response.

**Relation To Broader Scientific Literature:**

This paper improves the efficiency and accuracy of drag-based image editing methods using samples from videos.

**Theoretical Claims:**

There is no theoretical claim.

---

> ### Author Rebuttal · Authors · 2025-03-31
>
> Thanks for the insightful feedback and appreciations in our work. Please find our response to your questions below.
>
> **Q1**: Architectural design similar to Wear-Any-Way.
>
> **A1**: Thank you for pointing this out. While our architectural design indeed draws inspiration from prior literature, including Wear-Any-Way, we respectfully emphasize several critical differences that clearly distinguish our work:
>
> 1) **Broader Scope:** Our approach addresses the general setting of drag-based editing on **arbitrary images**, whereas Wear-Any-Way specifically targets editing human clothing images. The difference in scope significantly impacts the underlying challenges and method pipeline design.
>
> 2) **Novelty in Video as Supervision Signal:** A core novelty of our paper is the insight that natural videos serve as a scalable and effective source of supervision for training general-purpose drag-based editing models. In contrast, Wear-Any-Way does not utilize video data and is restricted to image-based supervision within a narrow human-clothing domain. Due to our broader and more challenging scenario, our data pipeline also requires additional sophisticated steps (e.g., filtering camera movements, filtering frames with object key-points go out-of-bound, etc.) to accurately and scalably construct training pairs from diverse video data.
>
> 3) **Out-of-domain Generalization Insights:** As discussed in introduction (L80-L86) and illustrated in our empirical evaluation, our model notably generalizes to editing tasks that involve motions unseen in training (e.g., local deformation of rigid structures or objects). This observation provides novel insights into how video-based supervision can effectively enable image editing across broader, unseen domains, which is absent from Wear-Any-Way. For a more detailed discussion on this aspect, please see **A2** of "Response to Reviewer oUVx".
>
> 4) **Additional Novel Test-time Techniques:** Given the nature of our task and approach, we introduce additional novel test-time techniques not covered by Wear-Any-Way, such as Point-Following Classifier-Free Guidance (PF-CFG), CFG scale scheduler for PF-CFG, point augmentation, and sequential dragging. These innovations significantly improve performance and reliability for general-purpose drag editing.
>
> We will revise the paper to better highlight these contributions and clarify the novelty of our approach.
>
>
> **Q2**: The qualitative experiments compared to other methods are not very comprehensive.
>
> **A2**: Thank you for the suggestion. We will include additional qualitative comparisons in the updated version to further strengthen our empirical analysis. That said, we would like to respectfully clarify that the seven examples presented were carefully selected to be representative, spanning a diverse range of styles, objects, and editing intentions. Moreover, the quantitative results in Table 1 are based on DragBench, a widely adopted benchmark dataset with diverse samples, which provides statistically meaningful evidence of our method’s effectiveness.
>
>
>
> **Q3**: Example in Fig. 1\(c\) seems to be a forward-moving camera. Does the author keep such data during the filtering process?
>
> **A3**: Thank you for the question. We confirm that all videos, including the one in Figure 1\(c\), were filtered according to the criteria stated in the paper to ensure static camera movement. The example in Figure 1\(c\) may appear to involve a forward-moving camera, but it actually features a static camera observing a scene where syrup is being poured onto a surface. The apparent change in scale results from the growing size of the syrup pile, not from any camera motion. We will clarify this in the revised manuscript to avoid confusion.
>
>
> **Q4**: It is suggested that the author add a section at the end of the introduction to summarize the contributions of this paper.
>
> **A4**: Thanks for the suggestion. We will include a summary of contribution at the end of the introduction in our updated version.

---

> > ### Comment · Reviewer_qwJU · 2025-04-09
> >
> > My concerns have been well addressed, so I am willing to increase my rating to 3.

---

> > > ### Author Response · Authors · 2025-04-09
> > >
> > > We’re glad to hear that our rebuttal helped address your concerns. Thank you again for your thoughtful review and constructive feedback. We truly appreciate the time and effort you put into evaluating our work.

---

### Official Review · Reviewer_oUVx · 2025-03-12

**Overall Recommendation:** 3

**Summary:**

The paper presents a new approach, LightningDrag, for drag-style editing problem. LightningDrag features in significantly faster inference speed and good generalization. LightningDrag is trained by watching paired video frames. Finally, the authors showcase the performance of LightningDrag by comparing with other drag-style editing baselines.

## update after rebuttal
The rebuttal resolved my concerns. Please include discussions accordingly to the revised paper. I've increased my rating to weak accept.

**Claims And Evidence:**

I appreciate most of the claims from the authors, but there are some claims that may need further discussions:
- The training mechanism proposed by the paper seems to show a significant overlap with Magic Fixup [1], which also uses paired video frames for training. However, there is no discussion or comparison with it in the paper. It might be very helpful to discuss or show by experiments the main difference with Magic Fixup.
- It is mentioned in L080-L086 that LightningDrag generalizes well to out-of-domain editing even if certain transformations have deformations. However, there is no theoretical or empirical evidence for it. It would be helpful if the authors could show some explanation for this part.


[1] Alzayer, H., Xia, Z., Zhang, X., Shechtman, E., Huang, J.B., and Gharbi, M. Magic fixup: Streamlining photo editing by watching dynamic videos. arXiv preprint arXiv:2403.13044, 2024.

**Essential References Not Discussed:**

One of the contributions, learning from paired video frames, is somehow pretty similar to Magic Fixup as mentioned in "Claims And Evidence". However, it has not been well discussed in the paper.

**Experimental Designs Or Analyses:**

Authors compare with most of the previous drag-style editing methods, but there are some missing:
- [1] Alzayer, H., Xia, Z., Zhang, X., Shechtman, E., Huang, J.B., and Gharbi, M. Magic fixup: Streamlining photo editing by watching dynamic videos. arXiv preprint arXiv:2403.13044, 2024.
- [2] Mou, Chong, et al. "Dragondiffusion: Enabling drag-style manipulation on diffusion models." ICLR 2024.
- [3] Jiang, Ziqi, Zhen Wang, and Long Chen. "CLIPDrag: Combining Text-based and Drag-based Instructions for Image Editing." ICLR 2025.

Also, there's no ablation study in the paper. It might be helpful to include an ablation study to show the effectiveness of each component.

**Methods And Evaluation Criteria:**

The method part generally looks good to me. The authors design the model based on Stable Diffusion (SD)'s inpainting model, conditioned on features from IP-Adapter, appearance & reference image features from the additional appearance encoder, point control from point embedding. It might be a plus to visualize these feature maps for a clear understanding of the model, but it is not necessary.

The evaluation criteria also look good to me. The authors follow DragBench and use IF and MD as evaluation metrics. The time efficiency is evaluated by runtime under the same experiment settings.

**Other Comments Or Suggestions:**

N/A

**Other Strengths And Weaknesses:**

Other strengths:
- The paper shows a promising result in fast drag-style editing. The quality of the end product is competitive to state-of-the-art methods.
- LightningDrag shows a good appearance preservation ability compared to other methods, as shown in Figure 7.
- The test-time strategies provide further refinement for users if needed.

Other weaknesses: N/A.

**Questions For Authors:**

Please see my reviews above. My major concern for this paper is its similarity to Magic Fixup. More discussions or evaluations would be beneficial to differentiate two papers.

**Relation To Broader Scientific Literature:**

N/A

**Theoretical Claims:**

There is no theoretical claim mentioned in the paper. This paper is application-driven.

---

> ### Author Rebuttal · Authors · 2025-03-31
>
> Thanks for the insightful feedback and appreciations in our work. Please find our response to your questions below.
>
> **Q1**: Need discussion on differences with Magic Fixup
>
> **A1**: We acknowledge that our approach shares with Magic Fixup the general idea of leveraging video data; however, we respectfully highlight several key differences:
>
> 1) **Editing Instruction**:
> Magic Fixup relies on user-provided "coarse edits", typically involving rough object translation or duplication. In contrast, our method employs drag-based editing, allowing users to specify precise points and exact movement trajectories. Thus, the editing intentions we target fundamentally differ from Magic Fixup. This further leads to difference in architetural design choice, where our architecture directly inject the encoded point embedding into self-attention modules for precise spatial control, while Magic Fixup place editing instruction at the input of the model.
>
> 2) **Complex Editing Capability**:
> Magic Fixup excels at edits such as object translation/movement, but might struggle with intricate transformations involving rotations with complex local details. For example, in Figure 7 (row 3), the editing intention of significantly rotating a face from side-view to frontal-view would be challenging to deliver using coarse edits. On the other hand, our drag-based approach clearly conveys and effectively executes such complex editing intentions.
>
> 3) **Data Construction Pipeline**:
> Our data preparation process differs significantly from Magic Fixup, involving precise point-tracking using state-of-the-art methods (e.g., CoTracker-2) and stringent filtering for static-camera videos. This pipeline provides accurate, scalable data specifically suited for general drag-based editing.
>
> 4) **Novel Test-Time Techniques**:
> We further introduce novel test-time strategies, including Point-Following Classifier-Free Guidance (PF-CFG), dynamic scale scheduling for PF-CFG, point augmentation, and sequential dragging. These innovations enhance precision and robustness, which are absent from Magic Fixup.
>
> 5) **Simplified Real-World User Interface**:
> Magic Fixup’s user interface inherently requires complex modules (e.g., segmentation models, rotation tools, zooming functions), making the interaction complicated. Conversely, our approach simplifies user interaction to just point-clicking and mask-painting, enabling intuitive editing and better user experience.
>
> We will incorporate these clarifications into our revised manuscript.
>
> **Q2**: Need more explanation for out-of-domain generalization edits.
>
> **A2**: We would like to clarify our claim with both empirical evidence and further explanation.
>
> Empirically, we demonstrate generalization to out-of-domain editing with deformation through a range of qualitative examples in Figures 3 to 7. In addition, our quantitative results are based on DragBench, which includes many samples involving local deformations (e.g., lengthening an object or deforming parts of a rigid structure). The strong performance on this benchmark supports our claim.
>
> Conceptually, we view this generalization as a form of compositional generalization. While our training videos do not contain explicit editing instructions such as stretching hair or lifting mountains, they do include non-rigid deformations --- such as in Figure 1\(c\), where syrup accumulates and changes shape over time. By learning to model such non-rigid transformations, the model is able to generalize to similar deformation behaviors even in unseen or rigid-object scenarios.
>
> Furthermore, our proposed Point-Following Classifier-Free Guidance (PF-CFG) plays a critical role in enabling this behavior. As shown in Figure 4, PF-CFG helps ensure that the semantic content around handle points is faithfully dragged toward the target, improving control and robustness. The benefit of PF-CFG is further confirmed in our ablation study (see **A1** of "Response to Reviewer RdPN").
>
> We will incorporate these points into the revised manuscript to better support and clarify our claim.
>
> **Q3**: Missing comparisons of CLIPDrag, DragonDiffusion, MagicFixup.
>
> **A3**: Since our work is concurrent to CLIPDrag, we did not include it in this version of submission. We will update their results in the updated version.
>
> As for the DragonDiffusion, we have compared with its improved version, which is DiffEditor. In addition, we have discussed DragonDiffusion in our introduction and related work. We will include the results of DragonDiffusion for sake of completeness in our updated version.
>
> As for MagicFixup, since it follows a very different editing instruction, it might be challenging to fairly conduct large-scale comparison with our approach. We will include qualitative comparisons with MagicFixup in our updated version and update the detailed discussion among these two approaches.
>
> **Q4**: Lacking in quantitative ablation studies.
>
> **A4**: Please see **A1** of "Response to Reviewer RdPN"

---

### Official Review · Reviewer_RdPN · 2025-03-14

**Overall Recommendation:** 3

**Summary:**

This paper presents Lightningdrag, a diffusion model trained on video data, enabling accurate and consistent drag-based edits within seconds, leveraging source noise prior and point-following classifier-free guidance for improved accuracy and consistency.

**Claims And Evidence:**

1. Lightningdrag formulates drag-based image editing as a diffusion-based approach trained on video data, incorporating user-specified points via point embedding attention, which is novel.

2. The paper proposes novel strategies, including test-time refinements using noised source latents and inverse square classifier-free guidance, which significantly improve the quality of editing results.

3. As demonstrated in the supplementary videos and Table 2, Lightningdrag achieves notable improvements in time efficiency, completing edits within approximately one second, a substantial advantage over previous methods.

**Essential References Not Discussed:**

No

**Experimental Designs Or Analyses:**

Overall, the experiment is extensive and thorough.Authors provide the detailed experimental settings and extra quantitate results in the supplementary.

**Methods And Evaluation Criteria:**

1. The proposed methods and evaluation criteria make sense for the targeted application. Table 2 and supplementary videos effectively demonstrate the advantages of the method in terms of efficiency and editing quality, clearly outperforming previous approaches.

2. Figures and qualitative results illustrate the effectiveness of source noise prior, inverse square classifier-free guidance, and sequential dragging. However, the paper lacks quantitative ablation studies for each individual technical contribution, limiting detailed insights into their respective impacts.

**Other Comments Or Suggestions:**

No

**Other Strengths And Weaknesses:**

**Strengths:**
- The presented approach achieves clear improvements in editing quality and efficiency, outperforming prior latent-diffusion and GAN-based methods.

**Weaknesses:**
1. Although the methods introduce beneficial components such as source noise prior and point-following classifier-free guidance, the paper lacks quantitative ablation studies, limiting the understanding of each component’s individual contribution.
2. The reason behind the significantly improved inference speed (as shown in Table 2) is not sufficiently explained, leaving the underlying factors contributing to high efficiency unclear.

**Questions For Authors:**

See weakness

**Relation To Broader Scientific Literature:**

Already discussed in the paper.

**Theoretical Claims:**

No Theoretical Claims

---

> ### Author Rebuttal · Authors · 2025-03-31
>
> Thanks for the insightful feedback and appreciations in our work. Please find our response to your questions below.
>
> **Q1**: Lacking in quantitative ablation studies.
>
> **A1**: Thanks for the advice. We conduct the detailed quantitative ablation study as you suggested. Results are given below.
>
> To start with, we conduct an ablation study to understand the effect of point-following classifier-free-guidance (PF-CFG).
>
> ||No CFG|Constant CFG|Square CFG|Linear CFG|Inv. Square CFG|
> |----|-----|-----|-----|-----|-----|
> |MD(&darr;)|22.27|17.76|17.73|18.59|18.95|
> |IF(&uarr;)|0.9071|0.8486|0.8743|0.8829|0.8896|
>
> In the above table, the first column is the results without PF-CFG, while the rest are results with PF-CFG under different CFG scale schedules. Comparing the first column with the second columns, we show that adding PF-CFG will significantly enhance point-following (better MD) while compromise the overall appearance preservation (worse IF). As we explore different CFG scale schedules (from square to inv. square, the CFG value decreases increasingly faster throughout denoising process), we show that decaying CFG values during denoising process can effectively reduce the degradation on IF. Among these schedulers, we select inv. square as our default CFG scale scheduler to attain the best overall appearance preservation while maintain good point-following. Our numerical results here align well with the visual illustration in Figure 4.
>
>
> Further more, we conduct ablation study on the choice of noise prior.
> ||Pure Noise|Mixed Noise Latents|Copy and Paste|Noise Source Latents|
> |----|-----|-----|-----|-----|
> |MD(&darr;)|19.76|19.01|18.53|18.95|
> |IF(&uarr;)|0.8878|0.8861|0.8876|0.8896|
>
> From the results, one can observe that "Noise Source Latents" and "Copy and Paste" can outperform the other two strategies. Comparing to "copy and paste", "noise source latents" have slightly better IF and slightly worse MD. Therefore, it is hard to discern which one is better among these two. However, since "copy and paste" involves much more complicated implementation, we select "Noise Source Latents" as our default strategy in the spirit of simplicity. Moreover, the numerical results presented here also align well with the visual illustration in Figure 3.
>
> **Q2**: The reason behind the improved inference speed is not sufficiently explained.
>
> **A2**: Thank you for pointing this out. The significant speedup arises from our reformulation of drag-based image editing as a conditional generation task, as mentioned in the abstract and introduction. Unlike prior methods such as DragDiffusion or DiffEditor, which rely on iterative latent optimization or gradient-based guidance (sometimes requiring up to 80 iterations of forward and backward passes through the diffusion UNet), our method avoids this costly process. Instead, we achieve drag-based editing via a single feed-forward pass on our conditional-diffusion-model, making the latency comparable to standard diffusion image generation (around 1~2 second). We will clarify and include this detailed explanation in the revised version of the paper.

---

### Official Review · Reviewer_4Wjk · 2025-03-15

**Overall Recommendation:** 3

**Summary:**

This paper proposes LightningDrag, a feed-forward diffusion model designed to achieve drag-based image editing. LightningDrag employs a point embedding network to encode drag instructions from users and preserves the image identity using an ID adapter and appearance encoder. Experiments demonstrate that LightningDrag achieves superior drag-based editing results without the need for slow optimization.

**Claims And Evidence:**

Yes

**Essential References Not Discussed:**

NA

**Experimental Designs Or Analyses:**

Yes

**Methods And Evaluation Criteria:**

Yes

**Other Comments Or Suggestions:**

NA

**Other Strengths And Weaknesses:**

Strengths:

1. The idea of injecting drag instructions as conditions into an inpainting diffusion model is well-founded. This approach eliminates the need for heavy latent optimization, resulting in significantly faster editing speeds.

2. The authors explore various noise initialization strategies and implement point-following classifier-free guidance to enhance performance.

3. LightningDrag achieves much faster inference speeds compared to existing methods.

4. The presentation is clear and easy to follow.

Weaknesses:

1. The overall framework involves fine-tuning an inpainting diffusion model with additional conditions. Both the ID-preserving attention and point-following attention techniques are borrowed from previous work. Additionally, LightningDrag encodes dragging conditions as full-resolution image features, which is redundant since most values are set to zero. Although the authors implement several optimizations to accelerate the diffusion process, some module designs remain sub-optimal.

2. The authors mention additional operations, such as point augmentation, but more analysis and experiments are needed. For example, what is the success ratio without these augmentations?

3. SDEDrag outperforms LightningDrag on DragBench. While the authors claim that SDEDrag often results in undesired identity mapping, it would be beneficial to conduct a user study to evaluate this claim more thoroughly.

**Questions For Authors:**

NA

**Relation To Broader Scientific Literature:**

Accelerate drag-based image editing.

**Theoretical Claims:**

Yes

---

> ### Author Rebuttal · Authors · 2025-03-31
>
> Thanks for the insightful feedback and appreciations in our work. Please find our response to your questions below.
>
> **Q1**: Limited novelties in architectural design.
>
> **A1**: As discussed in the related work section, although we take inspiration from some previous literature for architecture design, we are targeting a more general setting: drag-based editing on **general images**, rather than specific domains like faces or objects.
>
> One of the key novelties of our work is the insight that natural videos can serve as a viable and scalable training signal for general drag-based image editing. Moreover, as discussed in the introduction (e.g., Page 1 L80 to L86) and supported by empirical results (e.g., Table 1 and Figure 7), our approach generalizes well to out-of-domain manipulations that are not explicitly observed during training (e.g., hair lengthening, furniture deformation). This demonstrates a new perspective on leveraging video-based supervision for image editing with conditional diffusion models (See **A2** of Response to Reviewer oUVx for more discussion).
>
> In addition, our training data construction pipeline can inspire following work on how to preprocess video frames to train image editing models. Also, our framework introduces several novel test-time techniques, including point-following classifier-free guidance (PF-CFG), point augmentation, and sequential dragging, which improve controllability and performance. Finally, we will release our code and models to support open research.
>
> We will revise the paper to better highlight these contributions and clarify the novelty of our approach.
>
>
>
> **Q2**: Encoding dragging conditions as full-resolution features might be redundant.
>
> **A2**: For the dragging conditions, we would like to clarify that the point-embedding modules are relatively lightweight, accounting for only about 2% of the total model parameters (40M out of 2B). As such, encoding the point conditions as full-resolution feature maps does not introduce significant computational or memory overhead. More importantly, this design choice provides a clear spatial inductive bias, allowing the model to more accurately interpret and follow user-specified dragging instructions. We found this spatial encoding to be beneficial for learning precise and controllable drag-based editing behaviors. We will include this discussion in the updated version to better explain the motivation behind our design.
>
>
>
> **Q3**: More analysis are needed for "Drag Engineering" operations. What is the success rate without them?
>
> **A3**: All evaluation results --- both qualitative (Figures 7 and 8) and quantitative (Table 1) --- are obtained **without** using additional techniques such as point augmentation or sequential dragging. This demonstrates that our approach is robust and effective in many common cases without requiring extra operations. The additional techniques are designed to handle more challenging scenarios. Based on our experience with the DragBench dataset, our model produces satisfactory results on 162 out of 205 test samples without any additional operations. With point augmentation and sequential dragging, we are able to improve editing results on 21 more samples. We will clarify this in the revised version for better transparency.
>
>
>
> **Q4**: Need more clarification on results of SDEDrag.
>
> **A4**: A robust drag-based editing approach must achieve strong performance on **both** Image Fidelity (IF) and Mean Distance (MD). IF evaluates how well the overall appearance is preserved after editing, while MD measures how accurately the handle points are moved to their target positions. Although SDEDrag achieves a slightly better IF score, its MD (45.779) is significantly worse than ours (18.95), suggesting that it often fails to complete the intended dragging operation and instead produces results that remain close to the original image. This is also evident in the qualitative comparisons in Figure 7, where SDEDrag introduces minimal changes without effectively moving the handle points. We will include additional qualitative examples in the supplementary material to further support this point.

---

### Decision · Program_Chairs · 2025-05-01

**Decision:**

Accept (poster)

**Comment:**

This work initially received mixed reviews and the major concerns were around novelty, comparison to previous methods and ablation studies. The authors addressed the reviewers' concerns well during the rebuttal and a positive consensus has been achieved among reviewers.

I therefore recommend acceptance of the paper and strongly encourage the authors to integrate the rebuttal into the revision and to rewrite it as necessary to clarify their contribution.